# Clinical characteristics, management, and outcomes of pediatric non-infectious uveitis: A multicenter study in Saudi Arabia

Mohammed Nashawi[1,2*], Suzan A. Alshareef[3], Salma Alswealh[3], Nidhal Almohammady[3], Hamza Fida[3], Huda Ahmedhussain[4], Emtenan Basahl[3], Saddiq Habiballah[1,2], Abdulaziz Alshehri[5], Abdulelah BinYami[3], Sirin Alfaqih[6], Mahmoud Majeed[5], Manal Hadrawi[7]

1 Department of Pediatric, Faculty of Medicine, King Abdulaziz University, Jeddah, Saudi Arabia, 2 Immunology Unit, King Fahd Research Medical Centre, King Abdulaziz University, Jeddah, Saudi Arabia, 3 Faculty of Medicine, King Abdulaziz University, Jeddah, Saudi Arabia, 4 Department of Ophthalmology, Faculty of Medicine, King Abdulaziz University, Jeddah, Saudi Arabia, 5 Department of Pediatric, King Fahad Armed Forced Hospital, Jeddah, Saudi Arabia, 6 Department of Internal Medicine, Dr. Erfan & Bagedo General Hospital, Jeddah, Saudi Arabia, 7 Department of Opthalmology, King Fahad Armed Forced Hospital, Jeddah, Saudi Arabia

* manashawi@kau.edu.sa

## Abstract

### Background

Pediatric non-infectious uveitis is a challenging condition with significant risks for long-term ocular complications and visual impairment. Limited regional data on its clinical characteristics and outcomes in Saudi Arabia are available.

### Objective

This study aimed to describe the clinical features, management strategies, and outcomes of pediatric non-infectious uveitis in two major medical centers in Saudi Arabia.

### Methods

A retrospective observational study was conducted, including 36 pediatric patients diagnosed with non-infectious uveitis from January 2017 to December 2023. Data were collected on demographic characteristics, clinical presentation, laboratory findings, treatment modalities, and follow-up outcomes at 3, 6, 12, and 24 months. Descriptive and inferential statistics were used for analysis.

### Results

Among the 36 patients, 66.67% were female, and the mean age at diagnosis was 8.99±4.17 years. The most common etiology was juvenile idiopathic arthritis (52.8%), followed by idiopathic uveitis (25%) and Vogt-Koyanagi-Harada disease (19.4%).

**Data availability statement:** All relevant data are within the paper and its Supporting information files.

**Funding:** The author(s) received no specific funding for this work.

**Competing interests:** The authors have declared that no competing interests exist.

Bilateral involvement was present in 55.6% of cases, with anterior uveitis being the predominant type (75%). Antinuclear antibody (ANA) positivity was observed in 69.44% of patients. The most frequently used treatments at diagnosis included prednisolone acetate 1% (91.67%) and methotrexate (50%). Over the 2-year follow-up, 86.11% of patients achieved remission, although complications such as synechiae (13.89%), cataracts (8.3%), and band keratopathy (5.56%) were observed. Visual acuity outcomes improved in most cases, although delayed presentation and non-compliance posed challenges.

## Conclusion

Pediatric non-infectious uveitis in Saudi Arabia mirrors global patterns, with JIA being the leading cause. Early diagnosis and biologic therapies have improved remission rates and reduced complications. Future efforts should focus on enhancing screening, patient adherence, and access to advanced therapies to optimize outcomes for affected children.

## Introduction

Uveitis is an inflammatory condition of the eye that encompasses a variety of etiologies and clinical manifestations, posing a significant burden on pediatric populations. Although less common than in adults, pediatric uveitis accounts for 2–14% of all uveitis cases and often presents unique diagnostic and therapeutic challenges [1,2]. Non-infectious uveitis, particularly associated with systemic conditions like juvenile idiopathic arthritis (JIA), represents the most frequent cause in children, as highlighted in studies conducted globally and in Saudi Arabia [3,4].

The diagnosis of pediatric uveitis can be particularly challenging due to its often-insidious onset and asymptomatic progression in early stages. This silent nature can lead to delayed detection until complications such as cataracts, glaucoma, or visual impairment occur [2,5,6]. Additionally, the limited ability of young children to articulate symptoms further complicates timely diagnosis, making systematic screening crucial in high-risk groups, such as children with ANA-positive JIA [3,6].

Epidemiological data on pediatric uveitis in the Middle East, including Saudi Arabia, remains sparse. The limited studies indicate that idiopathic and JIA-associated uveitis predominates in this region, mirroring global trends [2,4]. These insights underscore region-specific research's importance in informing clinical management and improving outcomes.

Advances in managing pediatric uveitis have significantly improved outcomes over the past decade. The introduction of biologics, such as TNF inhibitors (e.g., adalimumab), has revolutionized the treatment of refractory cases, providing an effective alternative to traditional immunosuppressants and reducing reliance on corticosteroids [7–9]. Early and aggressive treatment strategies aim to prevent complications and preserve vision, emphasizing the importance of early diagnosis and intervention [6,10–12].

This study assesses the clinical features, complications, and treatment outcomes of pediatric non-infectious uveitis in Saudi Arabia. By analyzing data from two tertiary centers, the study seeks to contribute to the growing body of evidence on the region's epidemiology and management of pediatric uveitis. Insights from this research are expected to guide future clinical practice and policy development in Saudi Arabia and surrounding countries.

## Subjects and methods

Study design, setting, and time frame: A retrospective observational study was done at two centers, King Abdulaziz University Hospital and King Fahd Armed Forces Hospital Jeddah, Saudi Arabia, Patients diagnosed between January 2017 and December 2022 were included in the study. Data extraction and statistical analysis were conducted during 2023.

### Study participants

The inclusion criteria were patients diagnosed with non-infectious uveitis < 18 years of age. Eligible patients were identified through hospital electronic medical records and ophthalmology clinic registries using diagnostic codes related to uveitis. We excluded patients diagnosed with uveitis due to infectious causes. A total of 36 patients were included. Patients with incomplete follow-up data were included in analyses for the available follow-up intervals

### Definitions of clinical outcomes

Remission was defined as the absence of active intraocular inflammation for at least three months during follow-up. Recurrence was defined as the reappearance of inflammatory activity after a period of remission. Persistent uveitis was defined as continuous inflammatory activity without achieving remission during follow-up. The diagnosis and classification of uveitis were based on the Standardization of Uveitis Nomenclature (SUN) Working Group criteria. The grading of anterior chamber inflammation was performed according to the SUN grading system. Visual acuity was assessed using age-appropriate methods, including Snellen charts in cooperative children and fixation-based assessment methods in younger patients.

### Data collection

A pre-designed checklist was prepared to collect data about clinical features, laboratory workup, and treatment of these patients at the time of diagnosis (Table 1), three (Table 2), six (Table 3), twelve (Table 4), and twenty-four months after diagnosis (Table 4).

### Ethical considerations

The Biomedical Ethics Committee of King Abdulaziz University approved the study. Reference No 93 1–23

### Data analysis

Data were analyzed using the Statistical Package for Social Sciences (SPSS), version 26 (IBM Corp., Chicago, IL, USA). Descriptive statistics were used to summarize the data. Categorical variables were presented as frequencies and percentages, while continuous variables were reported as mean ± standard deviation. Patients with incomplete follow-up data were included in analyses for the available follow-up time points only.

## Results

The study included patients with pediatric non-infectious uveitis, of which 66.67% were female and 33.33% were male. The mean age at diagnosis was 8.99 ± 4.17 years (range: 3–18 years). Juvenile idiopathic arthritis (JIA) was the most common etiology, accounting for 52.8% of cases, followed by idiopathic uveitis (25%), Vogt-Koyanagi-Harada disease (19.4%), and psoriatic JIA (2.8%). Most cases (55.6%) involved both eyes.

Table 1. The basic data of the patients at the time of diagnosis.

| Basic data | | Mean±SD OR N (%) |
|---|---|---|
| Gender | Male | 12(33.3%) |
| | Female | 24(66.7%) |
| Age at diagnosis of uveitis (years) | Mean ±SD | 8.99±4.17 |
| | Mini-Max | (3-18) |
| Etiology of uveitis | Idiopathic | 9(25%) |
| | JIA | 19(52.8%) |
| | Psoriatic JIA | 1(2.8%) |
| | Vogt-Koyaniagi-Harda | 7(19.4%) |
| Anatomical location of uveitis | Anterior | 27(75%) |
| | Intermediate | 2(5.6%) |
| | Pan uveitis | 7(19.4%) |
| Affected eye | Both eyes | 20(55.6%) |
| | Left eye | 9(25%) |
| | Right eye | 7(19.4%) |
| IOP (Right eye) | Mean ±SD | 14.11±4.45 |
| | Mini-Max | 8-22 |
| IOP (left eye) | Mean ±SD | 15.43±4.35 |
| | Mini-Max | 8-20 |
| Cells grade in the anterior chamber (Right eye) | Mean ±SD | +1.82±0.97 |
| | Mini-Max | +0.5-3 |
| Cells grade in the anterior chamber (Left eye) | Mean ±SD | +2.0±1.31 |
| | Mini-Max | +0.5-+4 |
| Time between systemic disease diagnosis and uveitis(month) | Mean ±SD | 24.09±36.74 |
| | Mini-Max | 0-132 |
| Time between uveitis onsite and systemic disease(month) | Mean ±SD | 1.33±4 |
| | Mini-Max | 0-12 |
| Present of complications upon diagnosis of uveitis | cataract | 3(8.3%) |
| | secondary glaucoma | 1(2.78%) |
| | band keratopathy | 2(5.56%) |
| | Anterior or posterior synechiae | 5(13.89%) |
| | macular edema | 1(2.78%) |

N.B.: SD = standard deviation.

The ophthalmological examination at diagnosis revealed a mean intraocular pressure (IOP) of 14.11±4.45 mmHg (range: 8–22 mmHg) in the right eye and 15.43±4.35 mmHg (range: 8–20 mmHg) in the left eye. The anterior chamber cell grade averaged +1.82±0.97 (range: +0.5–3) in the right eye and +2.0±1.31 (range: +0.5–4) in the left eye. Laboratory findings showed a high prevalence of antinuclear antibody (ANA) positivity (69.44%) among patients, with mean CRP and ESR values of 17.92±29.39 mg/L (range: 0–22) and 5.82±7.60 mm/h (range: 1–91), respectively. Visual acuity was assessed at baseline and during follow-up visits using age-appropriate assessment methods.

The mean time between systemic disease diagnosis and uveitis onset was 24.09±36.74 months (range: 0–132 months), whereas the mean time between uveitis onset and systemic disease diagnosis was 1.33±4 months (range: 0–12 months). At diagnosis, most patients with uveitis received topical prednisolone acetate 1% (91.67%), cyclopentolate

**Table 2. The patients' basic data at the follow-up interval [3 months after diagnosis].**

| Basic data | | Mean ±SD OR N (%) |
|---|---|---|
| Interval of follow-up [3M after diagnosis] | Yes | 28(77.7%) |
| Uveitis status | Active | 9(25%) |
| Cells grade in the anterior chamber (Right eye) | Mean ±SD | +0.71±0.27 |
| | Mini-Max | +0.5-+1 |
| Cells grade in the anterior chamber (Left eye) | Mean ±SD | +1.38±1.75 |
| | Mini-Max | +0.5-+4 |
| Is there any change in the systemic treatment? | Yes | 9(25%) |

N.B.: SD = standard deviation.

**Table 3. The patients' basic data at the follow-up interval [9 Months after diagnosis].**

| Basic data | | Mean ±SD OR N (%) |
|---|---|---|
| Interval of follow-up [9M after diagnosis] | Yes | 29(80.56%) |
| Uveitis status | Active | 11(30.56%) |
| Cells grade in the anterior chamber (Right eye) | Mean ±SD | +2.0±0.94 |
| | Mini-Max | +0.5-4 |
| Cells grade in the anterior chamber (left eye) | Mean ±SD | +2.07±1.43 |
| | Mini-Max | +1-4 |
| Is there any change in the systemic treatment? | Yes | 5(13.88%) |

N.B.: SD = standard deviation.

**Table 4. The patients' basic data at the follow-up interval [1 year after diagnosis].**

| Basic data | | Mean ±SD OR N (%) |
|---|---|---|
| Interval of follow-up [1 year after diagnosis] | Yes | 28(77.78%) |
| Uveitis status | Active | 11(30.56%) |
| Cells grade in the anterior chamber (Right eye) | Mean ±SD | +1.36±0.95 |
| | Mini-Max | +0.5-3 |
| Cells grade in the anterior chamber (left eye) | Mean ±SD | +1.36±0.98 |
| | Mini-Max | +1-3 |
| Is there any change in the systemic treatment? | Yes | 8(22.22%) |

N.B.: SD = standard deviation.

(36.11%), adalimumab (22.22%), and oral prednisolone (19.44%), while treatments for systemic disease primarily involved methotrexate (50%), prednisolone (25%), and adalimumab (22.22%) (Table 1). Over the 2-year follow-up period (Tables 2–5), a significant number of patients experienced persistent or recurrent uveitis, with most requiring an increase in local corticosteroid therapy rather than an escalation of systemic treatments. Complications observed included anterior or posterior synechiae (13.89%), cataracts (8.3%), band keratopathy (5.56%), macular edema (2.78%), and secondary glaucoma (2.78%).

**Table 5. The basic data of the patients at the interval of follow-up [2 years after diagnosis].**

| Basic data | | Mean ±SD OR N (%) |
|---|---|---|
| Interval of follow-up [2 years after diagnosis] | Yes | 20(55.56%) |
| Uveitis status | Active | 5(13.89%) |
| Cells grade in the anterior chamber (Right eye) | Mean ±SD | +2.0±1.51 |
| | Mini-Max | +0.5-2 |
| Cells grade in the anterior chamber (left eye) | Mean ±SD | +1.16±0.76 |
| | Mini-Max | +0.5-2 |
| Is there any change in the systemic treatment? | Yes | 3(8.33%) |

N.B.: SD = standard deviation.

## Discussion

Pediatric non-infectious uveitis presents significant challenges in diagnosis, management, and long-term outcomes, as highlighted by this study conducted in a pediatric population in Saudi Arabia. Juvenile idiopathic arthritis (JIA) was identified as the most common etiology (52.8%), followed by idiopathic causes (25%), aligning with global trends that emphasize JIA as a leading cause of pediatric uveitis [5,6]. Anterior uveitis predominated in 75% of cases, reflecting similarities with international cohorts [7,13–16]. The high rate of antinuclear antibody (ANA) positivity (69.44%) underscores its value as a biomarker for JIA-associated uveitis and the necessity of routine ANA testing in high-risk populations [5]. Treatment strategies predominantly included corticosteroids and immunomodulatory agents such as methotrexate and adalimumab, with biological therapies like TNF inhibitors proving effective in refractory cases [8,10,11,17,18]. Encouragingly, 86.11% of patients achieved remission after two years of treatment, a rate comparable or even slightly higher to global reports, highlighting the efficacy of current therapeutic approaches when systematically applied [8,9,17,18]. However, complications such as synechiae (13.89%), cataracts (8.33%), and band keratopathy (5.56%) occurred, although at relatively lower rates compared to historical cohorts, reflecting improvements in diagnosis and treatment [12–16]. Delayed diagnosis, often due to the asymptomatic nature of anterior uveitis, remains a critical challenge, emphasizing the need for systematic screening programs for ANA-positive JIA patients [5,13–16]. The relatively higher proportion of VKH disease observed in this cohort may reflect regional genetic predisposition and ethnic variations reported in Middle Eastern populations. Future research should explore the role of emerging therapies such as JAK inhibitors and IL-6 inhibitors and further incorporate biomarkers and advanced imaging techniques to refine diagnostic and therapeutic strategies [8–11,19,20]. These findings underscore the importance of timely intervention, tailored treatments, and continued efforts to optimize outcomes for pediatric uveitis patients. Biologic therapies, particularly TNF inhibitors such as adalimumab, were used in a subset of patients with refractory disease and appeared to contribute to improved disease control. However, given the retrospective design and relatively small sample size, these findings should be interpreted cautiously, and further prospective studies are required to evaluate their long-term effectiveness

## Limitations

This study has several limitations. First, the retrospective design may introduce information bias and limit the availability of complete follow-up data. Second, the relatively small sample size may affect the generalizability of the findings. Finally, Regional differences in access to specialized pediatric ophthalmology and rheumatology services may influence the diagnosis and management of pediatric uveitis. In some areas, delayed referral to tertiary care centers and limited availability of advanced biologic therapies may contribute to differences in disease outcomes compared with high-resource settings.

## Conclusion

This study reinforces the significance of JIA and idiopathic causes as leading etiologies of pediatric uveitis in Saudi Arabia, highlighting the efficacy of biologic therapies in achieving remission and minimizing complications. Continued efforts to improve early diagnosis, patient adherence, and access to advanced therapies will be essential in optimizing outcomes for children with uveitis. Moreover, encouraging a multidisciplinary approach as joint clinics to ensure proper management of the patients

## Supporting information

**S1 Data. Data sheet.**
(XLSX)

## Acknowledgments

**Declaration of AI assistance:** The authors declare that artificial intelligence-based linguistic tools were used solely to enhance the manuscript's clarity and quality of the English language. All research ideas, study design, statistical analyses, results, and discussions presented in this paper are entirely authentic and the original work of the authors. The use of AI was limited to linguistic refinement and did not contribute to the conceptualization or scientific content of the manuscript.

## Author contributions

**Conceptualization:** Mohammed Nashawi, Suzan A. Alshareef, Salma Alswealh, Nidhal Almohammady, Hamza Fida, Huda Ahmedhussain, Emtenan Basahl, Saddiq Habiballah, Abdulaziz Alshehri, Abdulelah BinYami, Sirin Alfaqih, Mahmoud Majeed, Manal Hadrawi.

**Data curation:** Suzan A. Alshareef, Salma Alswealh, Nidhal Almohammady, Hamza Fida, Huda Ahmedhussain, Emtenan Basahl, Saddiq Habiballah, Abdulaziz Alshehri, Abdulelah BinYami, Sirin Alfaqih, Mahmoud Majeed, Manal Hadrawi.

**Formal analysis:** Mohammed Nashawi, Suzan A. Alshareef.

**Methodology:** Mohammed Nashawi.

**Project administration:** Mohammed Nashawi.

**Writing – original draft:** Mohammed Nashawi, Suzan A. Alshareef, Salma Alswealh, Nidhal Almohammady, Hamza Fida, Huda Ahmedhussain, Emtenan Basahl, Saddiq Habiballah, Abdulaziz Alshehri, Abdulelah BinYami, Sirin Alfaqih, Mahmoud Majeed, Manal Hadrawi.

**Writing – review & editing:** Mohammed Nashawi, Suzan A. Alshareef, Salma Alswealh, Nidhal Almohammady, Hamza Fida, Huda Ahmedhussain, Emtenan Basahl, Saddiq Habiballah, Abdulaziz Alshehri, Abdulelah BinYami, Sirin Alfaqih, Mahmoud Majeed, Manal Hadrawi.

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
