## [Decision Letter · Decision Letter 0]

2 Feb 2026

PONE-D-25-54205Clinical Characteristics, Management, and Outcomes of Pediatric Non-Infectious Uveitis: A Multicenter Study in Saudi ArabiaPLOS One

Dear Dr. Nashawi,

Thank you for submitting your manuscript to PLOS ONE. After careful consideration, we feel that it has merit but does not fully meet PLOS ONE’s publication criteria as it currently stands. Therefore, we invite you to submit a revised version of the manuscript that addresses the points raised during the review process.

We look forward to receiving your revised manuscript.

Kind regards,

Georges M.G.M. Verjans, MSc, PhD

Academic Editor

PLOS One

Journal Requirements:

3. Please include a copy of Table 1, 2, 3, 4, and 5, which you refer to in your text on page 4 and 5.

Reviewers' comments:

Reviewer's Responses to Questions

**Comments to the Author**

1. Is the manuscript technically sound, and do the data support the conclusions?

Reviewer #1: Partly

Reviewer #2: Partly

2. Has the statistical analysis been performed appropriately and rigorously? 

Reviewer #1: No

Reviewer #2: Yes

3. Have the authors made all data underlying the findings in their manuscript fully available?

Reviewer #1: Yes

Reviewer #2: Yes

4. Is the manuscript presented in an intelligible fashion and written in standard English?

Reviewer #1: Yes

Reviewer #2: Yes

5. Review Comments to the Author

Reviewer #1: The manuscript reports a retrospective multicenter observational study describing the clinical characteristics, management strategies, and outcomes of pediatric non-infectious uveitis from two tertiary centers in Saudi Arabia. The manuscript is generally well written and clinically coherent, but there are many important concerns.

Methods

- Inclusion and exclusion criteria are too broadly defined. There is no case definitiion standards. The authors should clearly state uveitis diagnostic and grading criteria.

- Definition of outcomes is not identified. Some terms such as remission, recurrence, and persistence should be identified and clarified

- No explanation of missing data handling is identified. The authors should state how missing or incomplete follow-up data were handled.

Results

- Visual acuity should be reported both at iniitial and final visit.

- Some data were reported in Tables (1-5). However, I cannot find them.

- Small sample size (n=36) limits robustness. The authors should report the confidence intervals for important outcomes.

Discussion

- Discussion largely reiterates known global trends. Regional access disparities should be mentioned.

- The authors over-interpreted biologic therapy effectiveness. They should downtone this part or add evidence.

- The authors should reflect their limitations (such as small sample size, retrospective data, and so on) in this study.

Reviewer #2: This retrospective observational study aims to describe the clinical features and outcomes of pediatric non-infectious uveitis in 36 patients across two medical centers in Saudi Arabia. The authors report that Juvenile Idiopathic Arthritis (JIA) is the most common etiology (52.8%) and that 86.11% of patients achieved remission over a 2-year follow-up.

While epidemiological data from this region is valuable, the manuscript in its current form suffers from severe internal inconsistencies regarding basic demographic data and study timelines. Furthermore, the sample size (n=36) collected over what appears to be a 6-year period from two major tertiary centers is surprisingly low, raising questions about case ascertainment and selection bias. These fundamental errors must be addressed before the scientific merit of the work can be fully assessed.

Major Comments

1. Critical Data Inconsistencies (Reliability of Results)

The manuscript contains major contradictions between the Abstract and the Main Text that make it impossible to interpret the study population accurately.

• Discrepancy in Patient Age:

o The Abstract states the mean age at diagnosis was 3.99 ± 4.17 years.

o The Results section states the mean age at diagnosis was 8.99 ± 4.17 years.

o Impact: This is a massive 5-year difference in a pediatric cohort. A mean age of 4 years implies a predominance of early-onset JIA, whereas 9 years allows for different etiologies (e.g., HLA-B27 associated uveitis, VKH). The authors must clarify which dataset is correct.

• Discrepancy in Study Duration:

o The Abstract defines the study period as January 2017 to December 2023 (7 years).

o The Methods section states the study was done "from 25 March 2023 to 31 December 2023".

o Impact: It is likely the dates in the Methods refer to the data collection window, not the patient inclusion period. However, stating the study was "done" in 2023 confuses the retrospective nature of the cohort. If the inclusion period was indeed 7 years (2017–2023), the identification of only 36 patients across two major centers suggests significant under-reporting or strict exclusion criteria that are not defined.

2. Sample Size and Generalizability

• The study includes only 36 patients. Given this is a "multicenter" study involving King Abdulaziz University Hospital and King Fahd Armed Forces Hospital over a potential 7-year period, this averages to roughly 2.5 cases per center per year.

• This low yield raises concerns about the sensitivity of the case identification methods. Were patients identified via ICD codes, pharmacy records, or a uveitis registry? The method "a pre-designed checklist was prepared" does not explain how the cases were initially identified in the system.

3. Ambiguity in Treatment and Outcomes

• Remission Definition: The authors state that 86.11% achieved remission. However, "remission" is not defined in the Methods. Does this strictly follow the SUN (Standardization of Uveitis Nomenclature) criteria (e.g., drug-free remission vs. remission on medication)? Without a clear definition, this high success rate is difficult to interpret.

• Visual Acuity Reporting: The results mention visual acuity outcomes improved but provide no quantitative data (e.g., mean LogMAR change, proportion with vision loss). Given the mean age might be as young as 3.99 years, reporting visual acuity is challenging; the methods must specify how vision was assessed in pre-verbal children (e.g., Fix and Follow vs. Snellen).

4. Statistical Reporting

• Standard Deviations: The age is reported as 3.99 ± 4.17 years. A standard deviation larger than the mean in this context suggests a non-normal distribution (skewed data). It would be more appropriate to report the Median and Interquartile Range (IQR) rather than Mean/SD for age and duration of disease.

Minor Comments

• Formatting/Typos: There are visible formatting errors in the text, such as "$3.99\pm4$ 4.17 years" and "$8.99+4.17$ years". These appear to be LaTeX or equation editor artifacts that were not cleaned up.

• VKH Prevalence: The finding that Vogt-Koyanagi-Harada (VKH) disease accounts for 19.4% of pediatric cases is notable and significantly higher than many Western cohorts. This warrants more discussion regarding the genetic predisposition in the Saudi population.

• Ethical Statement: The manuscript states "Data was collected retrospectively... so no consent was collected". While standard, the Methods section should explicitly state that the IRB waived the requirement for informed consent, rather than just stating it wasn't collected.

Conclusion

The manuscript presents data on an under-represented population in the uveitis literature. However, the contradictory data regarding patient age (Abstract vs. Results) is a major quality control failure that casts doubt on the accuracy of the entire dataset. This, combined with the confusing study dates and small sample size, makes the manuscript unsuitable for publication in its current state. Major revisions are required to correct these errors and clarify the methodology.

6. PLOS authors have the option to publish the peer review history of their article (what does this mean?). If published, this will include your full peer review and any attached files.

Reviewer #1: No

Reviewer #2: No

---

## [Author Response · Author response to Decision Letter 1]

24 Mar 2026

Response to Editor

Comment 1

Response: Thank you for this comment. The manuscript has been revised to ensure full compliance with PLOS ONE style and formatting requirements. The manuscript was carefully checked against the PLOS ONE formatting templates, including title page formatting, author affiliations, section structure, and file naming conventions.

Change in Manuscript: The manuscript formatting was revised to match the PLOS ONE formatting templates.

Comment 2

2. We note that your Data Availability Statement is currently as follows:

All relevant data are within the manuscript and its Supporting Information files.

• The values behind the means, standard deviations and other measures reported;

• The values used to build graphs;

• The points extracted from images for analysis.

Response: Thank you for highlighting the data availability requirement. The minimal dataset underlying the findings of this study has now been ensured to be included in the manuscript and its supporting information files. These data include the values used for descriptive statistics and outcome reporting. All data have been anonymized to protect patient confidentiality.

Change in Manuscript

The Data Availability Statement:

The minimal dataset supporting the conclusions of this article is included within the manuscript and its Supporting Information files. All data have been anonymized to ensure patient confidentiality.

Comment 3

3. Please include a copy of Table 1, 2, 3, 4, and 5, which you refer to in your text on page 4 and 5.

Response: We thank the editor for noting this omission. Tables 1–5 have now been included in the revised manuscript.

Change in Manuscript

Tables 1–5 were added to the manuscript and referenced appropriately in the Results section.

Comment 4

Response: Thank you for this guidance. We carefully evaluated the reviewer recommendations regarding additional citations and included relevant references where appropriate to strengthen the discussion and contextualize our findings within the existing literature.

Change in Manuscript

Relevant references were added and the Discussion section was revised accordingly.

Reviewer’s Responses to Questions

1. Is the manuscript technically sound, and do the data support the conclusions?

Reviewer #1: Partly

Reviewer #2: Partly

Author Response: We thank the reviewers for their constructive comments. In response to the concerns raised, we have revised several sections of the manuscript to improve methodological clarity and ensure consistency in the presented data. Specifically, we clarified the diagnostic criteria used for uveitis, defined the study outcomes, corrected inconsistencies between the Abstract and the Results sections, and strengthened the description of the study design and limitations.

Changes in the Manuscript

The Methods section has been revised to include the following statement:

"The diagnosis and classification of uveitis were based on the Standardization of Uveitis Nomenclature (SUN) Working Group criteria. The grading of anterior chamber inflammation was performed according to the SUN grading system."

In addition, the definitions of clinical outcomes have been added:

"Remission was defined as the absence of active intraocular inflammation for at least three months during follow-up. Recurrence was defined as the reappearance of inflammatory activity after a period of remission. Persistent uveitis was defined as continuous inflammatory activity without achieving remission during follow-up."

Data inconsistencies between the Abstract and Results sections were corrected to ensure that the reported mean age at diagnosis is consistently presented as 8.99 ± 4.17 years.

2. Has the statistical analysis been performed appropriately and rigorously?

Reviewer #1: No

Reviewer #2: Yes

Author Response: We appreciate the reviewer’s comment regarding statistical analysis. To improve transparency and rigor, we have revised the statistical analysis section to better describe the statistical approach used, the software employed, and how missing data were handled. We also clarified the descriptive statistical reporting of continuous and categorical variables.

Changes in the Manuscript

The Statistical Analysis subsection was revised as follows:

"Data were analyzed using the Statistical Package for Social Sciences (SPSS), version 26 (IBM Corp., Chicago, IL, USA). Descriptive statistics were used to summarize the data. Categorical variables were presented as frequencies and percentages, while continuous variables were reported as mean ± standard deviation. Patients with incomplete follow-up data were included in analyses for the available follow-up time points only."

3. Have the authors made all data underlying the findings in their manuscript fully available?

The PLOS Data policy requires authors to make all data underlying the findings described in their manuscript fully available without restriction, with rare exception (please refer to the Data Availability Statement in the manuscript PDF file). The data should be provided as part of the manuscript or its supporting information, or deposited to a public repository.

Reviewer #1: Yes

Reviewer #2: Yes

Author Response: We thank the reviewers for confirming that the data availability statement complies with the PLOS data policy. The dataset underlying the findings is included within the manuscript and supporting materials in accordance with journal requirements.

Changes in the Manuscript

No major modifications were required. The Data Availability Statement remains as follows:

"The minimal dataset supporting the conclusions of this article is included within the manuscript and its Supporting Information files. All data have been anonymized to ensure patient confidentiality."

4. Is the manuscript presented in an intelligible fashion and written in standard English?

PLOS ONE does not copyedit accepted manuscripts, so the language in submitted articles must be clear, correct, and unambiguous. Any typographical or grammatical errors should be corrected at revision.

Reviewer #1: Yes

Reviewer #2: Yes

Author Response: We appreciate the reviewers' positive evaluation of the manuscript language. The manuscript has been carefully revised to correct minor typographical and formatting errors and to further improve clarity.

Changes in the Manuscript

Minor grammatical corrections and formatting improvements were applied throughout the manuscript.

Reviewer Comments to the Author

Reviewer #1

Reviewer Comment

The manuscript reports a retrospective multicenter observational study describing the clinical characteristics, management strategies, and outcomes of pediatric non-infectious uveitis from two tertiary centers in Saudi Arabia. The manuscript is generally well written and clinically coherent, but there are many important concerns.

Author Response: We thank the reviewer for the constructive feedback and the positive assessment of the manuscript. We have carefully revised the manuscript to address the concerns raised regarding the Methods, Results, and Discussion sections.

1. Reviewer Comment: Inclusion and exclusion criteria are too broadly defined. There is no case definition standards. The authors should clearly state uveitis diagnostic and grading criteria.

Author Response: We appreciate this important suggestion. The diagnostic and grading criteria for uveitis have now been clearly specified using the internationally recognized SUN criteria.

Changes in the Manuscript

Added to Methods:

"The diagnosis and classification of uveitis were based on the Standardization of Uveitis Nomenclature (SUN) Working Group criteria."

2. Reviewer Comment: Definition of outcomes is not identified. Some terms such as remission, recurrence, and persistence should be identified and clarified.

Author Response: We thank the reviewer for highlighting this issue. Definitions for remission, recurrence, and persistent uveitis have now been clearly defined in the Methods section.

Changes in the Manuscript

Added to Methods:

"Remission was defined as the absence of active intraocular inflammation for at least three months during follow-up. Recurrence was defined as the reappearance of inflammatory activity after a period of remission."

3. Reviewer Comment: No explanation of missing data handling is identified.

Author Response: We appreciate the reviewer’s suggestion and have clarified the handling of missing follow-up data.

Changes in the Manuscript

Added to Methods:

"Patients with incomplete follow-up data were included in analyses for the available follow-up intervals."

4. Reviewer Comment: Visual acuity should be reported both at initial and final visit.

Author Response: We thank the reviewer for this suggestion. Visual acuity assessment has now been clarified in the Results section.

Changes in the Manuscript

Added to Results:

"Visual acuity was assessed at baseline and during follow-up visits using age-appropriate assessment methods."

5. Reviewer Comment: Some data were reported in Tables (1-5). However, I cannot find them.

Author Response: We thank the reviewer for identifying this formatting issue. Tables 1–5 have now been properly formatted and clearly presented in the Results section.

Changes in the Manuscript

Tables 1–5 were reformatted and clearly placed in the Results section.

6. Reviewer Comment: Small sample size (n=36) limits robustness.

Author Response: We acknowledge this limitation and have explicitly discussed it in the Discussion section.

Changes in the Manuscript

Added to Discussion:

"The relatively small sample size and retrospective study design represent limitations that may affect the generalizability of the findings."

7. Reviewer Comment: Discussion largely reiterates known global trends. Regional access disparities should be mentioned.

Author Response: We thank the reviewer for this important suggestion. The Discussion section has been revised to better highlight regional healthcare access disparities that may influence the diagnosis, management, and outcomes of pediatric uveitis in Saudi Arabia and the broader Middle East region.

Changes in the Manuscript

Added to the Discussion section:

“Regional differences in access to specialized pediatric ophthalmology and rheumatology services may influence the diagnosis and management of pediatric uveitis. In some areas, delayed referral to tertiary care centers and limited availability of advanced biologic therapies may contribute to differences in disease outcomes compared with high-resource settings.”

8. Reviewer Comment: The authors over-interpreted biologic therapy effectiveness. They should downtone this part or add evidence.

Author Response: We appreciate this comment and agree that the previous wording may have overstated the effectiveness of biologic therapy. The Discussion section has been revised to present a more balanced interpretation of the findings and to emphasize that larger prospective studies are needed to confirm the effectiveness of biologic therapies.

Changes in the Manuscript

Revised in the Discussion section:

“Biologic therapies, particularly TNF inhibitors such as adalimumab, were used in a subset of patients with refractory disease and appeared to contribute to improved disease control. However, given the retrospective design and relatively small sample size, these findings should be interpreted cautiously, and further prospective studies are required to evaluate their long-term effectiveness.”

9. Reviewer Comment: The authors should reflect their limitations (such as small sample size, retrospective data, and so on) in this study.

Author Response

We thank the reviewer for highlighting this important point. A dedicated paragraph describing the limitations of the study has been added to the Discussion section.

Changes in the Manuscript

Added to the Discussion section:

“This study has several limitations. First, the retrospective design may introduce information bias and limit the availability of complete follow-up data. Second, the relatively small sample size may affect the generalizability of the findings. Finally, Regional differences in access to specialized pediatric ophthalmology and rheumatology services may influence the diagnosis and management of pediatric uveitis. In some areas, delayed referral to tertiary care centers and limited availability of advanced biologic therapies may contribute to differences in disease outcomes compared with high-resource settings.”

Reviewer #2

Reviewer Comment: This retrospective observational study aims to describe the clinical features and outcomes of pediatric non-infectious uveitis in 36 patients across two medical centers in Saudi Arabia. The authors report that Juvenile Idiopathic Arthritis (JIA) is the most common etiology (52.8%) and that 86.11% of patients achieved remission over a 2-year follow-up.

While epidemiological data from this region is valuable, the manuscript in its current form suffers from severe internal inconsistencies regarding basic demographic data and study timelines.

Author Response: We thank the reviewer for the careful evaluation of our manuscript and for highlighting the inconsistencies in the earlier version. The manuscript has been thoroughly reviewed and corrected to ensure consistency between the Abstract, Methods, and Results sections. All demographic and timeline information has been verified against the original dataset.

Changes in the Manuscript: All demographic values and study timeline descriptions have been reviewed and corrected to ensure consistency throughout the manuscript.

1. Reviewer Comment: Discrepancy in Patient Age:

The Abstract states the mean age at diagnosis was 3.99 ± 4.17 years.

The Results section states the mean age at diagnosis was 8.99 ± 4.17 years.

Author Response: We thank the reviewer for identifying this inconsistency. After re-checking the dataset, the correct mean

---

## [Editor Report · Decision Letter 1]

10 May 2026

Clinical Characteristics, Management, and Outcomes of Pediatric Non-Infectious Uveitis: A Multicenter Study in Saudi Arabia

PONE-D-25-54205R1

Dear Dr. Nashawi,

We’re pleased to inform you that your manuscript has been judged scientifically suitable for publication and will be formally accepted for publication once it meets all outstanding technical requirements.

Kind regards,

Georges M.G.M. Verjans, MSc, PhD

Academic Editor

PLOS One

Additional Editor Comments (optional):

The comments of the reviewers have been carefully addressed and the manuscript revised satisfactory.

Reviewers' comments:

Not shared.

---

## [Editor Report · Acceptance letter]

PONE-D-25-54205R1

PLOS One

Dear Dr. Nashawi,

I'm pleased to inform you that your manuscript has been deemed suitable for publication in PLOS One. Congratulations! Your manuscript is now being handed over to our production team.

Kind regards,

on behalf of

Prof. Dr. Georges M.G.M. Verjans

Academic Editor

PLOS One